# Neurodevelopmental Aspects and Cortical Auditory Maturation in Children with Cochlear Implants

**DOI:** 10.3390/medicina56070344

**Published:** 2020-07-13

**Authors:** Cristina Pantelemon, Violeta Necula, Alexandra-Stefania Berghe, Livia Livinț-Popa, Steluța Palade, Vitalie Văcăraș, Ioana Anamaria Mureșanu, Ștefan Strilciuc, Fior-Dafin Mureșanu

**Affiliations:** 1Department of Neurosciences, “Iuliu Hatieganu” University of Medicine and Pharmacy, 400486 Cluj-Napoca, Romania; cristina_pantelemon@yahoo.com (C.P.); livia.popa@ssnn.ro (L.L.-P.); vvacaras@umfcluj.ro (V.V.); ioana@ssnn.ro (I.A.M.); dafinm@ssnn.ro (F.D.M.); 2“RoNeuro” Institute for Neurological Research and Diagnostic, 400364 Cluj-Napoca, Romania; 3Department of ENT, “Iuliu Hatieganu” University of Medicine and Pharmacy, 400486 Cluj-Napoca, Romania; neculav@yahoo.com; 4Department of Medical Informatics and Biostatistics, “Iuliu Hatieganu” University of Medicine and Pharmacy Cluj-Napoca, 400012 Cluj-Napoca, Romania; berghe.alexandra@gmail.com; 5Department of Pediatric Neurology, Children’s Emergency Hospital Cluj-Napoca, 400378 Cluj-Napoca, Romania; stelutaneuro@yahoo.com

**Keywords:** cortical auditory evoked potentials (CAEPSs), cochlear implantation, central auditory pathways, hearing loss, general development

## Abstract

*Background and objectives*: The cochlear implant is not only meant to restore auditory function, but it also has a series of benefits on the psychomotor development and on the maturation of central auditory pathways. In this study, with the help of neuropsychological tests and cortical auditory potentials (CAEPs), we intend to identify a series of instruments that allow us to monitor children with a cochlear implant, and later on, to admit them into an individualized rehabilitation program. *Materials and methods*: This is a longitudinal study containing 17 subjects (6 boys and 11 girls) diagnosed with congenital sensorineural hearing loss. The average age for cochlear implantation in our cohort is 22 months old. Each child was tested before the cochlear implantation, tested again 3 months after the implant, and then 6 months after the implant. To test the general development, we used the Denver Developmental Screening Test (DDST II). CAEPs were recorded to assess the maturation of central auditory pathways. *Results*: The results showed there was progress in both general development and language development, with a significant statistical difference between the overall DQ (developmental quotient) and language DQ before the cochlear implantation and three and six months later, respectively. Similarly, CAEP measurements revealed a decrease of positive-going component (P1) latency after cochlear implantation. *Conclusion*: CAEPs and neuropsychological tests prove to be useful instruments for monitoring the progress in patients with cochlear implants during the rehabilitation process.

## 1. Introduction

Hearing screening during the neonatal period helps with early identification of children with sensorineural hearing loss, opening access to a cochlear implant as a therapeutic option [1]. In the past, ahead of large-scale dissemination of neonatal hearing screening, the diagnosis of hearing loss would be established very late, around the age of 20–25 months old [2].

Sensorineural hearing loss is associated with late development, especially in the language domain [3,4,5]. At the same time, it affects motor, cognitive, and psychosocial domains as well [6,7,8,9]. Children diagnosed with deafness present motor development delay, mainly in coordination and balance, compared to healthy children of the same age [10]. It has also been observed that in the long term, in children with hearing loss, the late development of language affects neurocognitive processes, such as executive function, sequential processing, and concept formation [11]. Psychosocial difficulties and emotional impairments affect 20–50% of children with sensorineural hearing loss [12,13]. In a study carried on a sample of 334 children with sensorineural hearing loss, Dammeyer highlighted that the prevalence of social deficits among this group is 3.7 times higher compared to healthy children. The degree of impairment is higher in those with associated disabilities [14].

Cochlear implants are the elective treatment for children with severe and profound sensorineural hearing loss. An implant is a prosthetic device whose electrodes, placed at the level of the inner ear, directly stimulate the fibres of the auditory nerve; the signal is then transmitted to the auditory cortex and interpreted as auditory input [15].

The results of the cochlear implantation are influenced by several factors, such as the age of the cochlear implantation, the duration of hearing loss, the manner of communication, the bilateral implant, and the family’s engagement in the rehabilitation process [16,17].

Colletti et al. have shown that in children implanted before 12 months of age, auditory abilities, the development of language, and grammar skills have been significantly higher than in children implanted between 12 and 36 months of age. This increased rate of performance was maintained over ten years of study [18].

By restoring the auditory function with the cochlear implant, the child has the opportunity to develop normally.

The development of the auditory system occurs early on, in the intrauterine life, around the 25th week of pregnancy. Unlike the visual system (the visual function appears only after birth), the auditory system is refined by exposure to sounds starting from the 28th–30th week of pregnancy [19]. The early restoration of the auditory function allows the development of later cognitive skills, which depend on sensory experience [15]. The implantation before the age of 3.5–4.0 years allows a normal maturation of central auditory pathways [20,21].

The cortical auditory-evoked potential (CAEP) is a non-invasive method of examination and a reliable biomarker to quantify the development of central auditory pathways in normal-hearing and implanted children [22,23]. In children with normal hearing, CAEP morphology is governed by a positive peak known as the P1 component. In newborns, it has a latency of approximately 300 ms, followed by a rapid decrease during early childhood, reaching a latency of 125 ms at 3 years of age. Subsequently, it gradually reaches a latency of approximately 50–70 ms until adulthood [23,24]. The gradual decrease in P1 wave latency reflects an increase in the efficiency of transmitting sound along auditory pathways in the auditory cortex [25]. A study by Sharma et al. on a sample of 245 children with sensorineural hearing loss with cochlear implant suggests that children implanted before the age of 3.5 years old had a latency of the P1 wave within the normal limits, 6 months after implantation; half of those aged between 3.5 and 7 years old had normal latency, and those implanted after the age of 7 years old had delayed latency [21].

The plasticity of central auditory pathways reaches its peak during the first 3.5 years of life. It is then gradually reduced during the phenomenon of cross-reorganization, when the auditory areas develop visual and somatosensory functions [23,26,27]. Hearing loss has a series of consequences on the brain’s development that go beyond the impairment of the auditory system, even after the cochlear implantation. It can therefore be described as a disease of the connectome, where links are impaired not only within the auditory system but also in other sensory systems, with severe negative effects on high cognitive functions [15]. Early cochlear implantation during the sensitive period (i.e., when the brain’s plasticity is high), may prevent the development of degenerative alterations, facilitating appropriate functional development within the brain [28].

The current study aims to evaluate the benefit of cochlear implants on the psychomotor development and the maturation of central auditory pathways in children with sensorineural hearing loss. We hypothesize that early detection of a developmental delay allows for an early intervention with favourable long-term results.

## 2. Material and Methods

The study was approved by the Ethical Committee of Iuliu Hatieganu University of Medicine and Pharmacy of Cluj-Napoca, under process number 166 from 7 April 2017. For enrolling patients, we obtained informed consent of the child′s legal representative.

This study is a longitudinal clinical study. The criteria for inclusion in the study were children up to 6 years of age, diagnosed with severe and profound sensorineural deafness, using a cochlear implant, and included in an auditory–verbal rehabilitation program.

Each patient was assessed three times: before the cochlear implant, three months after the cochlear implant, and six months after the cochlear implant.

The development was tested using the Denver Developmental Screening Test II (DDST II), standardized to the Romanian population. The cortical auditory potentials were recorded to assess the effect of the cochlear implant on the maturation of central auditory pathways.

The Denver Developmental Screening Test (DDST II) is a test battery for children aged 0–6 years old, assessing the child’s general development. DDST II consists of 125 items that aim to evaluate the child in the following areas: personal–social (e.g., the interaction with other people, the ability to care for their own needs), fine motor-adaptive skills (e.g., hand-eye coordination, handling of small objects, solving problems), language skills (e.g., understanding and using language, combining words), and gross motor skills (e.g., walking, jumping, throwing ball overhead). Although DDST II was not initially used to yield a developmental quotient (DQ), recent studies have shown that the DQ obtained through DDST II may correlate to the diagnosis of late psychomotor development in children with complex medical conditions [29].

In the current study, DQ was used to assess the various parameters of development. The patient’s functional age was assessed using four parameters of development within DDST II. Overall DQ (expressed in percentages) was calculated using the Equation
DQ = estimated developmental functioning/chronological age × 100(1)

The DQ for assessing language (DQL) was calculated using the formula [29]:DQL = estimated developmental language functioning/chronological age × 100(2)

The CAEPs were recorded in a soundproofed room. During the procedure, the child was seated in a chair, and they watched cartoons without sound. Before the start of the procedure, the functionality of the cochlear implant was checked. The equipment used was Smart EP USB Jr. from Intelligent Hearing Systems (HIS 5020). (Intelligent Hearing Systems Co, Miami, FL, USA).

The International Electrode System 10–20 was used to place the electrodes on the scalp: the Cz active electrode was connected to the positive input of the amplifier, and the reference electrode was positioned on the mastoid contralateral to the cochlear implant, and connected to the negative input of the amplifier. The scalps of subjects were cleaned with abrasive gel. The electrodes were placed using a conductive adhesive. The level of impedance of the electrodes was maintained between 1–3 kΩ. The acoustic stimulus was delivered through a loudspeaker placed at a distance of 40 cm away from the implanted ear. The CAEPs were recorded in response to a synthesized speech syllable (“ba”) at an intensity of 70 dB nHL. The stimulus rate was 1.10/s, at a duration of 114,875 µsec for 512 sweeps, with the artefact rejection criterion at +/− 100 µV. Filter settings were 1–30 Hz, 100,000 gain, and the time window was between 0 ms pre-stimulus and 500 ms post-stimulus response. The waveforms recorded were analyzed in order to identify the P1 wave, defined as the first robust positive peak, with a latency between 50 and 300 ms.

The data collected were analyzed using SPSS v. 25 (IBM, United States). To see the differences between overall DQ and language DQ in all three assessments, we used a one-way repeated measures ANOVA. In order to consecutively compare the results of the three assessments, we showed the results of post-hoc tests. When comparing differences between latencies of *P1* wave medians during the three evaluation periods, we used the nonparametric, related samples, Wilcoxon signed-rank test due to data distribution. In particular, one sample of a Wilcoxon signed-rank test was used when comparing the median of one sample with zero (as in the beginning of the study, some of the characteristics of the patients were absent, therefore their measurement resulting in a null value). We also calculated Cohen’s *d* coefficient to report the size effect, using the G*power application. Analyses used a 0.05 alpha level for statistical significance. Effect sizes were calculated the size effect at 80% statistical power.

## 3. Results

We had 17 patients in our sample, six boys and eleven girls, with ages between 12 and 45 months old. The duration of deafness was between 9 and 43 months. Regarding the aetiology of sensorineural hearing loss, one child had a congenital cytomegalovirus infection, in one child the cause was genetic, and 15 had unknown causes. The characteristics of the studied sample are described in Table 1 below.

In the one-way repeated measures ANOVA model for overall DQ, we found statistically significant differences between the means (shown in Table 2) of the three evaluations using Greenhouse–Geisser correction (*F* = 8.530, *p* = 0.004, partial eta squared = 0.348). We used this correction because the sphericity assumption was violated (Mauchly’s test of sphericity; *p* = 0.008). The assumption of normality of the residuals was checked, and there is no evidence that it was violated. There were independent observations within each group.

The post-hoc tests showed statistically significant results, as seen in Table 3.

For language DQ, by applying the one-way repeated ANOVA test, we found statistically significant differences between the means (shown in Table 4) of the three evaluations (Wilks Lambda 0.259: *F* = 21.428, *p* < 0.001, multivariate partial eta squared = 0.741). There was no violation of test assumptions—as the observations were independent within the group, the normality of residuals assumption was not violated, nor was the sphericity (Mauchly’s test of sphericity: *p* = 0.114).

Post-hoc tests showed results as seen in Table 5.

The medians and interquartile ranges of latency of P1 are present in Table 6. The medians and interquartile ranges of latencies decrease over time. The comparison between the P1 peak latency during the three points in time shows a significant difference from a statistical viewpoint (*p* < 0.05). A significant difference in latency of wave P1 was observed after cochlear implantation. Despite the low sample size and reduced statistical power, large significant differences (Cohen’s *d* > 1.2) between the first and the second evaluation, as well as between the second and third, were observed.

## 4. Discussion

The objective of this study was to depict a neurodevelopmental profile (of psychomotor development) and to quantify the maturation of central auditory pathways in children with cochlear implants.

The majority of studies on children with cochlear implants have aimed to assess the expressive and receptive language, and not so much the cognitive abilities (general development), especially in children with early implantation. This study may offer a broader overview of the benefits of early cochlear implant at an early age.

In our approach, the results gathered pre- and post-implantation regarding the children’s performances in various growth areas show the progress in both general development and language development. We have observed a significant growth between the overall DQ before the implant and then 3 months later, post-implant, and again 6 months after the implant. These results are in accordance with those obtained by Paluch et al. in a study on three children using cochlear implants, who were implanted before the age of 2.5 years old. This group has shown that after nine months of using the cochlear implant, in the non-verbal cognitive domain, patients registered average or above-average scores [30]. It is important to notice that in our study, the average age for implantation was 22 months of age.

Hearing loss with onset in the first years of life has negative effects on intelligence development [31]. Due to auditory impairment, children with hearing loss recognize and process the outside information partially, which harms their cognitive development [32]. In a study on children with bilateral hearing loss, Meinzen-Derr et al. compared the results of psychomotor development before the age of 3, with a non-verbal IQ performed between the ages of 3–6 years old. The results showed that psychomotor development, mainly in the adaptive domain, can be a predictive factor for non-verbal cognitive measures. Therefore, patients who registered low scores in the first evaluation were left with lower functioning in the other evaluations as well, while those with higher cognitive functioning maintained their scores in the other evaluations [33].

The phonological environment during the first years of life affects language development and the functioning of cortical structures involved in language development. At the same time, the auditory cortex plays an important role in developing high cognitive functions [7].

Shin et al. showed in their study that after cochlear implantation, deaf children presented improvements in their cognitive abilities. Comprehension, concentration, sequential processing, and working memory were improved, reaching almost normal values. At the same time, their performance in information, comprehension, similarity, and mathematics was not significantly improved [32]. A cochlear implant not only has the role to restore auditory function, but it plays an important role in cognitive development [32].

The majority of existing studies focus on assessing the non-verbal cognitive abilities at pre-school age. Therefore, overall DQ can offer a useful baseline, helping to quantify the progress and identify delays that may appear later on, during the development age, in the case of children with early cochlear implantation.

On language development, although children from our study presented a delay in the language domain, we saw a significant statistical difference between the language DQ before the cochlear implant and the language DQ at 3 and 6 months after the implant.

The cochlear implant is meant to prevent delays in expressive and receptive language development [34]. As the implantation age is younger, the development of language is closer to that of children with normal hearing. In a study, Karltorp et al. showed that 4 years after implantation, children implanted until the age of 9 months had a language development similar to a normal-hearing child of the same age, while children implanted between 18 and 29 months old had delayed development of language [35]. In a study by Colletti et al. on a sample of 10 children who had been implanted before the age of 1, the authors showed that babbling appeared after 1–3 months after the implant activation [18].

At the same time, aside from early intervention, long-term monitoring is important for language development after the cochlear implant. In a study performed on children implanted between 5 and 18 months old, Wie et al. showed that, at four years after the implant, these patients obtained similar expressive and receptive language scores to the normal hearing population [5]. The delay in language acquisition in children with hearing loss later affects the learning process [3]. Receptive vocabulary is an important predictive factor in reading comprehension [36].

Another important predictive factor for language development in children with cochlear implant is the mother–child interaction and the family’s involvement in the auditor–verbal rehabilitation process. Children coming from families that are involved in the verbal rehabilitation process have significantly better results in language skills [3,4]. Also, a higher family income has been associated with better results regarding language abilities in children with cochlear implants [37,38].

For the quantification of the maturation of central auditory pathways, we used the P1 CAEP biomarker. In our study, together with the growth in the period of cochlear implant use, there is a significant decrease in the P1 wave latency. In the initial evaluation (pre-implant), there was no prominent response of the P1 component; later, however, after the implant, we can see a significant decrease in the component’s latency (Figure 1); the results are in concordance with the data in the literature [23,39,40].

The early age for the cochlear implantation, 22 months in our group, is an important factor that contributes to the significant decrease in the P1 latency. The cochlear implant guarantees an adequate stimulation of central auditory pathways, leading to an increase in the leading speed at a neuronal level by accelerating synaptogenesis and myelination [39,41]. The P1 wave latency can be a useful instrument of measurement of the efficiency of therapeutic intervention through cochlear implant.

The auditory cortex has a sensitive period of up to 3.5 years when its response to the auditory stimulation is at the maximum [21,23]. Performing the cochlear implant during this interval keeps a high plasticity in the central auditory pathways, and allows for the right maturation of the auditory cortex [21]. Dorman et al. showed that in children with cochlear implants, after 7 years, the P1 wave latency of auditory stimulation never reached the normal limits, even after more years of implant use [26]. These results demonstrate the use of the P1 biomarker in the evaluation of the maturation of central auditory pathways, and in the quantification of the results after the intervention in children with cochlear implants.

The relatively low number of subjects is attributable to procedural constraints; recording CAEPs at a young age is a limitation of our study. Future studies with a larger number of subjects are necessary to confirm the results in the present study. Nevertheless, the significance of our findings is important for the current literature, because these findings emphasize the importance of the assessment of general development and central auditory maturation in children with early cochlear implantation. With the use of neuropsychological tests, clinicians may identify deficits in all domains of development, not only in language development, therefore integrating care and providing patients access to an individualized rehabilitation program.

## 5. Conclusions

Cochlear implants have beneficial effects on both language development and other domains of development, as well as on the maturation of central auditory pathways. CAEP testing and the evaluation of psychomotor development are useful instruments in quantifying the long-term benefits of cochlear implants, and should be used in conjunction to provide the optimal patient-oriented therapeutic strategy.

## Figures and Tables

**Figure 1 medicina-56-00344-f001:**
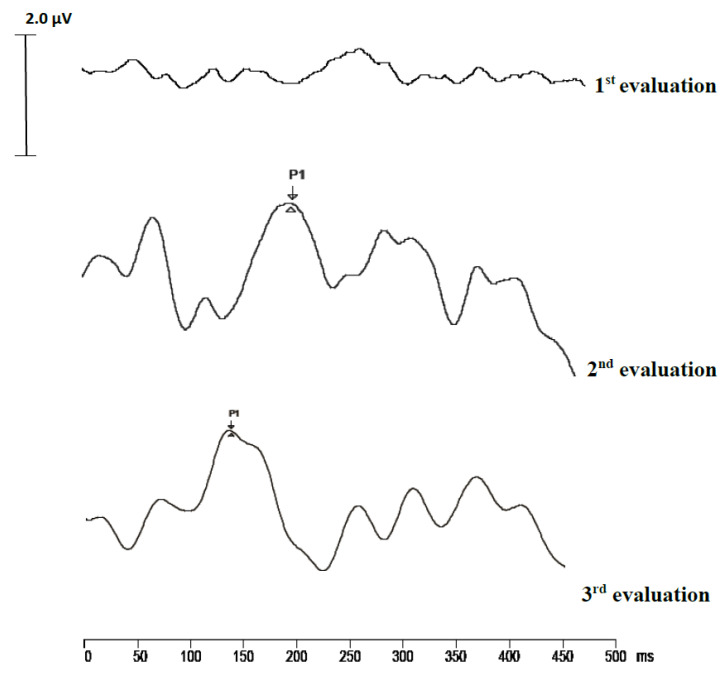
Cortical auditory potential (CAEP) trace recording in a child at three time points: pre-implantation, at three months after cochlear implant, and six months after cochlear implant.

**Table 1 medicina-56-00344-t001:** Description of the studied sample.

Variables	All Samples (*n* = 17)
Gender ^(a)^	-
Boys	6 (35.29%)
Girls	11 (64.71%)
Age (months) ^(c)^	First evaluation: 26.00 (9.89); second evaluation: 30.00 (9.46); third: evaluation 33.29 (9.21)
Duration of using hearing aid before cochlear implant (months) ^(b)^	5 [10]
Age of cochlear implantation ^(b,c)^	22 [16] ^(b)^ and 25.88 ± 9.90 ^(c)^

^(a)^ Numerical summaries are absolute (number) and relative frequencies (%). ^(b)^ Numerical summaries are median and interquartile range (IQR). ^(c)^ Numerical summaries are mean ± standard deviation.

**Table 2 medicina-56-00344-t002:** Comparative descriptive analysis of the overall developmental quotient (DQ) at the three evaluation moments.

Overall DQ	Mean (SD)
First evaluation	59.592 (11.242)
Second evaluation	64.554 (11.974)
Third evaluation	70.501 (10.641)

DQ, developmental quotient; SD, standard deviation.

**Table 3 medicina-56-00344-t003:** Comparative statistical analysis of overall DQ between the three evaluation moments.

First and second evaluation	*p* = 0.004
Second and third evaluation	*p* = 0.075
First and third evaluation	*p* = 0.002

**Table 4 medicina-56-00344-t004:** Comparative descriptive analysis of the language DQ at the three evaluation moments.

Language DQ	Mean (SD)
First evaluation	11.794 (5.185)
Second evaluation	20.770 (10.729)
Third evaluation	28.032 (10.267)

**Table 5 medicina-56-00344-t005:** Comparative statistical analysis of the language DQ between these three evaluation moments.

First and second evaluation	*p* = 0.001
Second and third evaluation	*p* = 0.054
First and third evaluation	*p* < 0.001

**Table 6 medicina-56-00344-t006:** Comparative statistical analysis of the latency of the P1 component at the three evaluation moments.

	Median ± IQR			*p*-Value	
	Initial	Second	Third	Initial vs. Second	Second vs. Third
Latency (ms)	0	175.00 ± 38.00	133.00 ± 69.80	<0.001 (*d* = 6.07)	<0.001 (*d* = 1.70)

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
