# Peer review of "Neurodevelopmental Aspects and Cortical Auditory Maturation in Children with Cochlear Implants"

_medicina, 2020, doi:10.3390/medicina56070344_

Round 1

Reviewer 1 Report

The study reported changes in two important parameters after cochlear implant in children, including functional development (DDST II) and maturation of central auditory pathways (CAEP P1 latency).  The results suggest that both measurement could be useful to monitor the rehabilitation process in patients after implantation.  The study was clear and straightforward, with elegant statistical analysis. There is no major flaw in the presentation, except some minor confusing statements:

Line 14: 3 and 6 months after the implant, respectively?

Line 17: define DQ on first use

Line 48 and 50: it is really awkward to use “age of 1” and “age of 12 months old” in the same sentence.

Line 83: Not sure what this means: “early detection of latency in development”

Line 165-166: It is unclear what was the comparison “between the electrophysiological evaluations during the three points in time”.  Is it the P1 peak latency?

Line: 223: … performed on children “implanted” between 5 and 18 months…. ?

Line: 223-226: the whole sentence is confusing, and needs to be revised for clarification.

Author Response

Point 1 – Line 14: 3 and 6 months after the implant, respectively?

Response 1: Each child was tested before the cochlear implantation, and tested again 3 months after the implant and then 6 months after the implant.

Point 2 – Line 17: define DQ on the first use

Response 2: developmental quotient (DQ)

Point 3 – Line 48 and 50: It is really awkward to use “age of 1” and “age of 12 months old” in the same sentence.

Response 3: Colletti et al. have shown that, in children implanted before 12 months of age , auditory abilities, the development of language and grammar skills have been significantly higher than in children implanted between 12 and 36 months of age.

Point 4 – Line 83: Not sure what this means: ”early detection of latency in development”

Response 4: that early detection of a  developmental delay

Point 5 – Line 165-166: It is unclear what was the comparison “between the electrophysiological evaluations during the three points in time”. Is it the P1 peak latency?

Response 5: The comparison between the P1 peak latency during the three points in time shows a significant difference from a statistical viewpoint (p<0.05).

Point 6 - Line 223:...performed on children “implanted” between 5 and 18 months…?

Response 6: In a study performed on children implanted between 5 and 18 months old,..

Point 7 – Line 223-226: the whole sentence is confusing, and needs to be revised for clarification.

Response 7: In a study performed on children implanted between 5 and 18 months old, Wie OS et al. showed that, at four years after the implant, these patients obtained similar expressive and receptive language scores to normal hearing population.

Reviewer 2 Report

The manuscript from Pantelemon C et al. reports on the feasibility of using neuropsychological tests and cortical auditory-evoked potential to monitor psychomotor development and maturation of the central auditory pathways following cochlear implantation in children. They have used the adapted Denver Developmental Screening Test II development quotient and CAEP P1 latency as the monitoring metrics. This reviewer has only minor criticisms:

  1. The authors have reported the overall DQ. The individual domain scores from the adapted DDST II have not been reported. There is a statement regarding delayed language development (Lines 211-12). The manuscript can be improved by including a table for the individual domain scores.
  2. Grammatical errors, poor word choice:

Line 13 – cohort is a better word than lot

Line 14 – incorrect use of “respectively”

Line 115 – contralateral is more commonly used

Line 123 – waveforms instead of pathways

Line 184 – incorrect use of “respectively”
